# Comparison of CML Rainfall Data against Rain Gauges and Disdrometers in a Mountainous Environment

**DOI:** 10.3390/s22093218

**Published:** 2022-04-22

**Authors:** Roberto Nebuloni, Greta Cazzaniga, Michele D’Amico, Cristina Deidda, Carlo De Michele

**Affiliations:** 1IEIIT, Consiglio Nazionale delle Ricerche, 20133 Milano, Italy; 2DICA, Politecnico di Milano, 20133 Milano, Italy; greta.cazzaniga@polimi.it (G.C.); cristina.deidda@polimi.it (C.D.); carlo.demichele@polimi.it (C.D.M.); 3DEIB, Politecnico di Milano, 20133 Milano, Italy; michele.damico@polimi.it

**Keywords:** commercial microwave links, disdrometers, rain gauges, rainfall sensors, rainfall, microwave propagation, rain attenuation

## Abstract

Despite the several sources of inaccuracy, commercial microwave links (CML) have been recently exploited to estimate the average rainfall intensity along the radio path from signal attenuation. Validating these measurements against “ground truth” from conventional rainfall sensors, as rain gauges, is a challenging issue due to the different spatial sampling involved. Here, we assess the performance of a network of CML as opportunistic rainfall sensors in a challenging mountainous environment located in Northern Italy. The benchmark dataset was provided by an operational network of rain gauges and by three disdrometers. Moreover, disdrometer data were used to establish an accurate relationship between path attenuation and rainfall intensity. A new method was developed for assessing CML: time series of rainfall occurrence and rainfall depth, representative of CML radio path, were derived from the nearby rain gauges and disdrometers and compared with the same quantities gathered from the CML. It turns out that, over the very short integration times considered (10 min), CML perform well in detecting rainfall, whereas quantitative rainfall estimates may have large discrepancies.

## 1. Introduction

Commercial microwave links (CML), widely used to interconnect cellular base stations, have been recently exploited as opportunistic sensors to estimate the average rainfall intensity along the radio path [1,2,3,4,5], and to reconstruct rainfall maps over a geographic region [6,7,8,9,10,11] or at country scale [12,13,14,15,16]. This technique is of particular interest when rain gauges and/or weather radars are not available [17]. The rainfall intensity is estimated from the time series of signal attenuation along the propagation path. Attenuation is obtained from the corresponding time series of transmitted signal level (TSL) and received signal level (RSL), that is, the raw data generated by the network management system that monitors link quality during CML operation.

CML networks are ubiquitous, and TSL and RSL data are often logged by their monitoring systems; hence, they have the potential to provide precipitation data at the global level. However, there are several sources of inaccuracy in the measurement of rainfall by CML. On the equipment side there are, among others, link downtime, occurrence of outliers, errors in raw data (e.g., signal quantization), and conversion into average values. On the environment side, the most frequent issues are: errors in the identification of rainy events, quantification of spurious components that affect path attenuation, such as the wet antenna contribution, spatial inhomogeneity of the rainfall field along the radio path, and errors in attenuation to rainfall intensity conversion due to variations in the drop size distribution (DSD). A survey of the issues associated with rainfall estimation by microwave link is presented in [18].

As far as the instrumental sources of inaccuracy are concerned, it should be noted that, during CML operation, only a limited amount of TSL and RSL data are stored. A common format for data storage is the minimum and maximum values of TSL and RSL every 15 min (the so-called MIN-MAX format). These values are often roughly quantized before being logged. The impact of quantization is larger for low values of attenuation. Hence, it depends on rainfall intensity, path length, and frequency, being larger for low rainfall intensities, short links, and low frequencies. The impact of data format, specifically MIN-MAX values within a time window of 15 min, was addressed in [4,19,20]. It was shown that the distribution of RSL in the above window is not symmetrical around the mean value, but it is skewed towards the minimum. Hence, the arithmetic average of the extremes overestimates the mean. To mitigate this problem, the aforementioned authors proposed unbiased estimators of the mean rainfall intensity: using a weighted average of MIN and MAX rainfall values, or correcting the MIN-MAX average by a multiplicative coefficient. The authors of [19] proposed a method to quantify the bias produced by the combined effect of quantization and MIN-MAX sampling.

Among the environmental sources of inaccuracy, a major problem is the correct classification of each time interval into wet or dry [21], since this process can have a dramatic impact on the estimation of the baseline, i.e., the RSL just before and after a rain event [22]. In fact, with a correct baseline estimation, it is possible to separate attenuation due to rain from signal variations due to unwanted effects, such as gas attenuation (typically water vapor), wet antenna loss, and electromagnetic ray bending due to variations in humidity, temperature, and pressure. The impact of wet antenna has been investigated by several authors [23,24,25]. Antenna wetting introduces extra attenuation that depends on the characteristics of the antenna (i.e., on the water-repellent properties of its radome), and it increases with frequency. Depending on the scenario, up to several dB of extra attenuation can be experienced during heavy rainfalls. This extra loss must be estimated and removed from total path attenuation. Another source of error may lie in the relationship used to convert attenuation into rainfall intensity [26], which is usually expressed in terms of a power law: (1)$$\gamma =k\cdot R^{\alpha }$$
where γ is the specific rain attenuation (dB km^−1^), *R* is the rainfall intensity (mm h^−1^), and *k* and α are coefficients dependent on wave frequency and polarization, on the path elevation angle and on the DSD. The DSD exhibits much natural variability from event to event and even within the same event [27]. Since *R* and γ have a very different analytical dependence on the DSD [26], different DSD correspond to different values of *k* and α: as a consequence, the same value of measured attenuation may correspond to different values of *R*. When no specific information on the DSD is available, it is customary to adopt the “standard” γ-*R* relation proposed by the ITU-R [28]. Finally, since Equation (Equation 1) is in general non-linear, spatial inhomogeneity of the rain field can introduce a bias. Hence, Equation (Equation 1) does not return the space-averaged rainfall intensity when the input is the space-averaged specific attenuation. The impact of the bias introduced by using Equation (Equation 1) is strongly dependent on link frequency.

The validation of the rainfall estimates obtained through CML is still an open issue, since ad hoc deployments of rainfall sensors are seldom feasible. Indeed, the “ground truth” is usually derived from preexisting meteorological sensors, typically operational rain gauge (RG) networks. In this respect, we should be aware of the different nature of CML and RG, which may result in different rainfall estimates. In particular, RG collect single-point measurements, whereas CML perform path-averaged measurements. Hence, if precipitation is not uniform along the CML path (e.g., due to orographic effects or occurrence of small rain cells with respect to the path length), there will be significant differences in rainfall estimates.

In this paper, we assess the performance of a network of CML as opportunistic rainfall sensors in the challenging mountainous environment of Valmalenco valley, in Northern Italy. The benchmark dataset consists of direct measurements collected by an operational network of RG and data from three disdrometers (DIS), deployed in the area to complete the test-bed. Data gathered from DIS were used to calibrate the γ-*R* relationship in Equation (Equation 1). Once calibration was performed, CML-derived rainfall estimates were compared with the measurements of the rainfall sensors. To that end, we envisaged a procedure that mitigates the effects of the different spatial sampling performed by CML and RG+DIS. The paper is structured as follows: In Section 2 we present the case study and describe the experimental setup. In Section 3, we describe the processing applied to CML and DIS data, together with the method for the calibration of the CML using DIS-derived DSD. In Section 4, we present the results obtained from the analysis of 15 rainfall events recorded during the experimental campaign. Finally, in Section 5 we draw the conclusions.

## 2. Case Study and Experimental Setup

The case study is the Valmalenco valley, an Alpine area in the province of Sondrio (Northern Italy). It is a narrow and steep valley crossed by Mallero river with altitudes ranging from 282 to 4018 m a.s.l., reached by Piz Bernina. The meteorological regime is seasonal with solid precipitation during the cold months and convective rain cells, associated with sudden drops in temperature, during summer. Figure 1 shows the study area. The positions and the altitudes of the rainfall sensors exploited in the present work are highlighted in different colors. The characteristics of the sensors are detailed in the following.

### 2.1. Commercial Microwave Links

The CML network, owned by Vodafone Italia S.p.A., includes 18 links. The links terminals are located at very different altitudes (from 280 to 2293 m a.s.l.) resulting in elevation angles up to 20°. Table 1 lists CML characteristics, namely, length, frequency of the available channels (two or four), altitude, path elevation angle and format of the raw data. The data stored by the network monitoring tool are the minimum and the maximum values of TSL and RSL, during consecutive non-overlapping 15 min time windows (15 min MIN-MAX format). For the purposes of this study, the data acquisition process of the four CML along the path of Valmalenco valley (links 9, 10, 11, and 13) was modified to store nearly instantaneous power measurements (one sample every 10 s). The data of the above four links were used for rainfall estimation, and the entire set of links was used for rainfall identification (Section 3.2).

### 2.2. Disdrometers

The disdrometers (DIS) used in this work are Thies Clima laser precipitation monitors (TLPM). They include a laser source and a transmitting optics, which generates an infrared light beam propagating through a very short atmospheric path. At the receiving end, the beam is focused by an optical lens onto a photo diode, which transforms the received optical power into an electric signal. The measurement principle relies on the decrease in the received signal when the infrared beam is crossed by a falling raindrop: the raindrop diameter is calculated from the depth of signal fade, whereas its velocity is derived from the duration of the fade. Besides particle diameter and velocity, DIS returns the precipitation intensity and the type of precipitation (rain, snow, hail, mixed precipitation). Three TLPM were deployed at three points in the Valmalenco valley located several kilometers apart from each other (see Figure 1) and at rather different altitudes (401, 648, and 1243 m a.s.l.). Each TLPM generated telegram files, in the form of ASCII text data, which were stored by a data logger and periodically transferred through the mobile network to a remote server.

### 2.3. Rain Gauges

Rain gauges are the most widespread sensors used for retrieval of liquid precipitation in meteorological and hydrological practice. Since Valmalenco valley is typically subjected to floods and landslides, it is equipped with a dense hydro-meteorological network managed by ARPA Lombardia (i.e., the regional agency for environmental protection) and by CMG (Sondrio’s centre of geological monitoring). The network includes ten tipping-bucket RGs with a 0.2 mm tip sensitivity. The RG data are available for download at the ARPA Lombardia website at three different integration times: 10 min, hourly, and daily. It is worth noticing that these sensors are non-heated; hence, they may be subjected to freezing when temperature goes below 0 °C. Therefore, when dealing with RG data, it is important to check whether the temperature is above freezing or not.

## 3. Methods

Methods to gather rainfall data from CML have been extensively discussed in the literature. Here we focus on a few specific aspects that are not often considered. First, disdrometers can be used to calibrate CML. In fact, by processing the particle counts collected by disdrometers, it is possible to predict microwave attenuation produced by a uniform layer of raindrops along the propagation path. Moreover, we propose an algorithm of identification of rainfall events based only on CML data, and a method for comparing rainfall estimates obtained from CML against direct measurements from conventional single-point rainfall sensors, i.e., rain gauges and disdrometers.

### 3.1. Disdrometer Data Processing

The raw data collected by each TLPM are particle counts over a 1 min observation time binned in a 2D histogram, including 20 velocity classes and 22 diameter classes. Velocity bins range from 0 to 10 m s^−1^, whereas diameter bins go from 0.125 to 8 mm. In addition, the TLPM returns estimates of liquid, solid, and total precipitation intensity *R* (mm h^−1^) every 1 min. The latter can be retrieved directly from the particle counts, given the effective measuring area of the sensor, by the following formula [29]: (2)$$R=6\pi 10^{-4}\sum_{i=1}^{N_{D}}\sum_{j=1}^{N_{V}}D_{i}^{3}\frac{n_{ij}}{A_{e}\Delta t}$$
where $D_{i}$ (mm) is the center of the *i*-th diameter bin, $n_{ij}$ is the number of droplets in the bin (i,j), *j* is the velocity bin index, $N_{V}$ is the number of velocity bins, $N_{D}$ is the number of diameter bins, $A_{e}$ (m^2^) is the effective sampling area, and Δt (s) is the sampling interval. Particle counts are used here to calculate rainfall attenuation (per unit path length) at a given rainfall intensity (see Section 3.3) by an electromagnetic model. As rainfall attenuation can be derived from CML raw data, the above relationship is crucial in gathering rainfall estimates from CML.

### 3.2. CML Data Processing

The raw data gathered by CML located in Valmalenco valley are the time series of TSL and RSL. The instantaneous power measurements collected every 10 s by links 9, 10, 11, and 13 of Figure 1 are indeed averages over four consecutive samples spaced 200 ms apart at a resolution of 1/8 dBm. The resulting average is subsequently quantized with a resolution Q=1 dB. Signal quantization is an important limitation of the CML in Valmalenco as it bounds both link sensitivity to light rainfall and rainfall measurement accuracy. Quantization is an additive random noise with uniform distribution between −Q/2 and +Q/2, which adds up to TSL and RSL. Let us assume that only RSL is significantly affected by quantization, as it is orders of magnitude lower than TSL, and that the quantization error propagates through the signal processing chain, affecting rain attenuation, again with an uniform distribution. Then, it is straightforward to calculate the minimum measurable rainfall intensity by a CML of length *L*:(3)$$R_{min}={\left(\frac{Q}{2L\kappa }\right)}^{\frac{1}{\alpha }}$$
where Q=1 dB in our case. The sensitivity of the 18 CML in Valmalenco ranges from 1 mm h^−1^ to about 2.7 mm h^−1^ (Figure 2a). The sensitivity of the CML used to estimate rainfall intensity (i.e., CML 9, 10, 11, and 13) is 2, 1.1, 2.1, or 1.3 mm h^−1^. Figure 2b shows the uncertainty bounds of rainfall estimates at the 95% confidence intervals for the above four links. In the worst cases (CML 9 and 11), the uncertainty can be as high as 20% even with intensities of 7–8 mm h^−1^. Please that the above figures refer to rainfall intensity estimates obtained from instantaneous RSL measurements. The integration time considered in this work is 10 min; that is, 60 CML samples were averaged. Assuming independent samples reduces the uncertainty by a factor close to 8. Moreover, CML transmit over one or two frequency channels either way; hence, they provide up to four measurements across a single path, which can be averaged.

Extracting rainfall estimates from CML data is not straightforward. Several authors investigated this topic, and there are a number of algorithms available [1,13,20,21,25,30]. CML data processing usually goes through the following basic steps:1.Classification of time intervals as dry or wet;2.Baseline (BL) calculation;3.Total path attenuation calculation;4.Calculation of the attenuation component due to wet antennas;5.Calculation of rainfall attenuation;6.Conversion of rainfall attenuation into rainfall intensity.

Dry/wet classification makes BL calculation more efficient and helps to remove artifacts from the time series of the rainfall intensity. Moreover, formulating a classification algorithm based on CML data permits one to assess their ability to detect rainfall. The algorithm proposed here is based on proximity: a CML is flagged as dry or wet during a certain period of time, by comparing the RSL time series with the ones of the neighbor links. This technique becomes more accurate when the number of links increases; hence, it was implemented over the larger set of 18 CML in Figure 1, which collect data in the 15 min MIN-MAX format. As a disadvantage, a 15 min time window is rather long, and it may include dry and wet intervals, especially in the presence of intermittent rainfall.

The dry/wet classification algorithm works as follows. CML *j* is neighbor of CML *i* if at least one of the following conditions hold: (a) *i* and *j* have one (and only one) terminal in the same position, (b) *i* and *j* intersect each other, (c) the average distance of *j* from *i* is within 2 km. Condition (c) is assessed by breaking CML *j* into 1 km segments and calculating the distance of each segment to CML *i*. If at least 50% of segments are within 2 km from CML *i*, then the neighborhood condition (c) is fulfilled. A reference RSL value is subsequently calculated for each CML. It quantifies the received signal during optimum propagation conditions, i.e., when only the attenuation component due to atmospheric gases is present. The observation period before a rainfall event is scanned by a sliding window (8 h long). The median value of MIN and MAX RSL are calculated in each window and the windows are subsequently sorted in descending order of MAX RSL. The reference RSL corresponds to the MAX RSL value in the first window where the MIN-MAX difference is within 1 dB, i.e., the quantization step. The classification of the 15 min time slot *t* of CML *i* starts by taking the difference between reference RSL and MIN of RSL in *t* for CML *i* and for the $N_{o,i}$ CML overlapped to *i* (e.g., CML 14 and 15 in Table 1) and for its $N_{n,i}$ neighbors. The above difference is thresholded by an hysteresis method based on lower and upper thresholds equal to 1 and 2 dB, respectively. The result is stored in the binary variable *B*; B=1 if the threshold is exceeded. The threshold values were chosen according, again, to the quantization step and to the estimate of the maximum expected attenuation in the absence of rain. For each of two CML sets, *i* and $N_{o,i}$ (set 1) and $N_{n,i}$ (set 2), the following indicator is calculated: (4)$$W_{i}\left(t\right)=\frac{\sum_{m=1}^{M}w_{m}B_{m}}{\sum_{m=1}^{M}w_{m}}$$
where $w_{m}$ is the reciprocal of the minimum rainfall intensity detectable by CML *m*. That is, the threshold outcome is weighted according to CML sensitivity. The above weights are corrected to take into account the effect of overlapped CML among the neighbors. If CML m≠i has already been classified, then the coefficient $B_{m}=1$ if *m* is wet, and $B_{m}=0$ if *m* is dry. If $W_{i,1}$ and $W_{i,2}$ are the coefficients relative to the above two sets, the classification rules are as in the following Table 2. Uncertain slots can be removed during offline processing according to the following rules: (i) uncertain slots between wet slots are wet, (ii) uncertain slots just before or after wet slots are wet. Finally, the algorithm can be made iterative, setting up an exit loop condition based on the difference in dry/wet classification with respect to the previous iteration.

The BL quantifies the average RSL in the absence of rainfall. The method used here is similar to the one in [30]. The BL is estimated by a sliding window of width $N_{w}$. If data are in the 15 min MIN-MAX format, the median value of the average between MIN and MAX RSL in every slot falling into $N_{w}$ is taken as the BL level in that window. The BL calculated with the sliding window method is valid only if all the 15 min slots in the window are dry and if the window starts at least $T_{g}$ seconds after the previous wet period to avoid the transition produced by antennas getting dry. To minimize the probability of BL contamination, it is assumed that $T_{g}=8$ h. In the dry periods where the BL value is invalid, according to the $T_{g}$ rule, but it is less than the one in the adjacent dry periods, the calculated BL value is flagged as valid. Finally, in the remaining periods of time (dry, wet or uncertain), the BL is calculated by interpolation.

Total path attenuation in a wet period is the difference between the BL level and the actual RSL. The attenuation due to antenna wetting is calculated using the two-parameter model outlined in [25] and subtracted from total attenuation to obtain the path-averaged rainfall attenuation component. The model parameters are the duration of the dry-wet transient, which we set equal to 15 min as in [25]; and the maximum wet antenna attenuation, here equal to 2 dB, that is, rather close to the 2.32 dB value found in [25] through an ad hoc experimental set-up and an optimization procedure. Finally, rain attenuation is divided by the path length and it is converted into path-averaged rainfall intensity by inverting Equation (Equation 1).

### 3.3. Calibration of the γ-*R* Relationship

The relationship between rain attenuation and rainfall intensity can be approximated by the power law function in Equation (Equation 1). The values of the coefficients *k* and α are tabulated in ITU-R [28], as functions of wave frequency and polarization. However, these coefficients are dependent on the microphysical properties of rainfall; hence, in principle, they are site- and event-dependent. When local DSD data are available, it is possible to retrieve *k* and α by regressing the specific rainfall attenuation, γ with the rainfall intensity, *R*. The value of γ (dB km^−1^) at a specific wave frequency *f* can be calculated from TLPM particle counts as follows: (5)$$\gamma =\frac{4.343\times 10^{-5}}{A_{e}\Delta t}\sum_{i=1}^{N_{D}}C\left(D_{i}\right)\sum_{j=1}^{N_{V}}\frac{n_{i,j}}{V_{j}}$$
where $A_{e}$, Δt, $n_{i,j}$, $N_{D}$, and $N_{V}$ are as in Equation (Equation 2); $V_{j}$ is the velocity (m s^−1^) of the *j*-th bin center; and $C(D_{i})$ is the extinction cross-section (in mm^2^) of a raindrop in the *i*-th diameter bin, which depends on the frequency and polarization of the incoming wave and on the incidence direction for non-spherical particles. Here raindrops are assumed oblate spheroids with an equivolume diameter equal to the center of the corresponding TLPM bin and an axial ratio given by [31]. In this case, $C(D_{i})$ can be calculated by numerical methods, e.g., by the Fredholm integral method [32]. As the path elevation angle can be significantly different from zero in the case of links located in a mountainous area, in principle, it is necessary to calculate $C(D_{i})$ at the correct elevation. In practice, it suffices to compute $C(D_{i})$ at horizontal and vertical polarization only, as κ and α in Equation (Equation 1) for any path geometry and wave polarization can be obtained from the ones at horizontal and vertical polarization [28]. The underlying assumption is that the power-law approximation holds for the rainfall dataset considered here, which is the case, as shown later in Section 4.2.

Power-law best fits have been worked out from 1 min estimates of *R* and γ, obtained through Equations (Equation 2) and (Equation 5), respectively, from the particle counts of the three TLPM. The optimum values of κ and α were obtained by linear least-square error minimization on log–log axes. Particles counts were classified as outliers and discarded if falling into velocity–diameter bins far from the expected V–D relationship for raindrops. Moreover, rainfall intensity values outside the interval 0.2–200 mm h^−1^ were filtered out. Please note that the quantity measured by CML is attenuation, that is, γ, whereas *R* is derived from γ. Hence, the correct way to proceed would be to calculate the optimized coefficients of the inverse of Equation (Equation 1). The results are very similar, though; hence, the two procedures are practically equivalent. Moreover, it is useful to quantify the goodness of fit on *R* rather than on γ axis. To this end, we used the standard RMSE (mm h^−1^) and the coefficient of variation, expressed as a percentage; that is,
(6)$$CV=100\frac{\sqrt{\frac{1}{N}\sum_{n=1}^{N}{\left(R_{n,dis}-\hat{c}\gamma_{n,dis}^{\hat{d}}\right)}^{2}}}{\frac{1}{N}\sum_{n=1}^{N}R_{n,dis}}$$
where *N* is the total number of observations, $\gamma_{n,dis}$ is the specific rainfall attenuation (dB km^−1^) from disdrometer data, and $\hat{c}={\hat{\kappa }}^{-1/\hat{\alpha }}$ and $\hat{d}=1/\hat{\alpha }$ are best fit coefficients.

Finally, the differences between the power-law best fits from different disdrometers and the ones between disdrometers and ITU-R relationship are quantified by a maximum relative difference in the corresponding *R* values as follows: (7)$$\Delta_{R}=100\max_{i}\left\{\frac{|R_{i,2}-R_{i,1}|}{0.5\left(R_{i,1}+R_{i,2}\right)}\right\}$$
where the subscripts 1 and 2 indicate different γ-*R* expressions (e.g., from two different disdrometers), and the index *i* indicates a point on the γ axis. That is, the γ axis is sampled in a number of points and the corresponding *R* values are calculated.

### 3.4. Validation of CML Rainfall Measurements

Data from CML networks can be interpolated into a spatial grid and validated against radar data [33]. However, in the case of Valmalenco, this approach is not practical due to two reasons. First, the number of CML is limited and their geometry is not favorable to carrying out a spatial reconstruction of the rainfall field. Moreover, the usage of radar data as a benchmark is questionable in a mountainous area due to beam-shielding and clutter contamination. On the other side, comparing the path-averaged rainfall estimates produced by individual CML against single-point measurements by conventional rainfall sensors is not straightforward when ad hoc test-beds are not available. In this section, we propose a procedure to assess the performances of individual CML as quantitative rainfall sensors against rain gauges and disdrometers. Specifically, we consider the dataset provided by the RG network owned by ARPA Lombardia, including ten sensors in the Valmalenco area (Figure 1), and the three TLPM deployed for this study. Please note that, in principle, the disdrometer dataset is not fully independent of CML data, as TLPM were used to calibrate the γ-*R* relationship.

The comparison between rainfall estimates from CML and from RG+DIS is based on the following procedure:1.Each CML is associated with a set of rainfall sensors (RG+DIS) according to a rule based on the distance;2.The time axes of CML, RG, and DIS data are synchronized and resampled at the scale of 10-min, i.e., the one of RG;3.A CML is flagged as dry or wet during a 10 min time slot, according to the status of the set of associated RG+DIS. If at least one sensor is wet, the time slot is flagged as wet. Moreover, CML return their own dry/wet binary time series according to the procedure in Section 3.2;4.The following quantities are calculated: contingency table for dry/wet classification and 10 min rainfall depth.

Ten minutes is the integration time of the network of available rain gauges in Valmalenco, which was the benchmark. To mitigate the effect of the different spatial sampling performed by CML and by RG+DIS, we generated synthetic time series of 10 min rainfall depth from the set of RG+DIS associated with each CML. A mean rainfall depth from RG+DIS was obtained by weighting each RG (or DIS) measurement according to an average distance to the CML, which was calculated as shown in Figure 3. The CML is broken into *N* segments, and an average RG-to-CML (or DIS-to-CML) distance is calculated, averaging the distances to the CML segments. The average rainfall depth estimate from the $M_{i}$ rainfall sensors close to CML *i* is a weighted average where the sensor *j* has the following weight: (8)$$W_{j}=\frac{\sum_{n=1}^{N}d_{j,n}}{\sum_{j=1}^{M_{i}}\sum_{n=1}^{N}d_{j,n}}$$
The above definition of distance takes into account both CML length and the relative position between CML and RG (DIS). Finally, we generated limiting time series by selecting MIN and MAX values from the RG+DIS set associated with each CML in every 10 min time slot. If RG+DIS cover the area around the CML path, it is expected that MIN and MAX time series are lower and upper bounds to CML-based estimates.

The difference between path-averaged and single-point measurements is amplified by the occurrence of highly non uniform rainfall. We quantify the degree of spatial homogeneity of rainfall by the variation among RG+DIS measurements. Specifically, if the number of rainfall sensors $\left(S_{i}\right)$ nearby CML *i* is limited (in our case it ranges from three to five), we consider the (normalized) difference (Δ) between the extremes, i.e.,
(9)$$\Delta_{ext}=\frac{\max_{S_{i}}R_{10}\left(j,t\right)-\min_{S_{i}}R_{10}\left(j,t\right)}{\frac{1}{S_{i}}\sum_{j=1}^{S_{i}}R_{10}\left(j,t\right)}$$
where the summation is done over the RG+DIS associated with CML *i*. If $\Delta_{ext}<T$, then rainfall is homogeneous.

### 3.5. Database of Events

Rainfall events were identified by analyzing the time series of 10 min rainfall depth collected by RG+DIS. An event has a maximum rainfall intensity in excess of 2 mm h^−1^, a duration of at least 30 min, and it is separated from the following event by at least 24 h of no precipitation. Hence, a rainfall event may include several episodes of rain. Overall, 15 precipitation events were detected during the observation period from July to October 2019. We considered only liquid precipitation events, that is, events during which the measured air temperature recorded by two thermometers located at SFF and LAP (see Figure 1) was well above 0 °C. The characteristics of the database of rainfall events are reported in Table 3—namely, the starting date and end date of the event, the number of rainfall episodes, and the duration of the event as number of rainy minutes. For each event are also reported the minimum and maximum cumulative rainfall depth recorded by the RG+DIS in the area (column 6) and the maximum rainfall intensity as detected by DIS (column 7).

## 4. Results

### 4.1. Verification of Disdrometer Data

Disdrometer data were used as ground truth and to calibrate the γ-*R* relationship in Equation (Equation 1). Hence, some verification in the TLPM dataset was necessary. A comparison between rainfall intensity values given by Equation (Equation 2) and the ones returned by the TLPM (not reported here) highlights that the agreement is usually good. The velocity of raindrops measured by the TLPM has been compared with Maitra and Gibbins’ formula for the terminal velocity of raindrops [34], which extrapolates the classic exponential model due to Atlas and Ulbrich to small particle sizes. Figure 4 shows the 2D histogram of raindrop diameter and terminal velocity for a moderate event and an heavy rainfall event. The best fit from data are close to Maitra and Gibbins’ formula. The TLPM curve sits slightly above the model at small diameter values during the heavy rainfall event in Figure 4b. Indeed, it has been shown that at high rainfall intensities, small raindrops may fall with larger velocities than would be expected from their diameters [35]. Please note that the calculation of γ in Equation (Equation 5) needs raindrop velocity data. To that end, we used the values measured by TLPM. Particle counts corresponding to velocity values lying at least 30% below the Maitra and Gibbins curve or at least 200% above it were discarded.

Last, we compared DIS and RG rainfall estimates, specifically CAG and SPR data; the sensors are located 1.36 km apart and are at the same altitude. The scatterplot of 1 h rainfall depth for all the events in the database is shown in Figure 5. The agreement is fairly good.

### 4.2. Optimization of *k* and α Coefficients

The γ-*R* pairs calculated from 1 min TLPM data through Equations (Equation 2) and (Equation 5) were fitted to the power-law model in Equation (Equation 1). The process was repeated for every event in the database and for every TLPM to check for any dependence of the α and κ coefficients on the location or on the event. The results are reported in Table 4. The specific attenuation γ was calculated at 18.8 GHz with vertical polarization and assuming an horizontal path. The power-law fit exhibits CV values usually within 20% for the 15 different events. The Pearson correlation coefficient (not shown) is always larger than 0.90. The dependence on the location was quantified through the $\Delta_{R}$ indicator of Equation (Equation 7) sampling the rainfall axis between 0.2 and 200 mm h^−1^ (last column of the table). When three TLPM are available, a more general expression of Equation (Equation 7) is used, considering three sets of *R* values instead of two. The dependence of the γ-*R* relationship on the location is rather limited ($\Delta_{R}$ usually within 25%). Best fits at different frequencies in the Ka band (17–23 GHz) return similar numbers. Results in the Ku band (where only CML 10 operates) are slightly worse. Finally, power-law best fits from aggregation of TLPM data highlight differences from one event to another at small rainfall intensities. $\Delta_{R}$ is within 30% if restricted to the interval 5–200 mm h^−1^.

Figure 6 shows the best fit power-law curve on log-log axes, putting together all the events and all the disdrometers (black line), and the ITU-R curve (red line) [28]. The γ-*R* pairs derived from 1 min disdrometer counts are shown as well in three different colors. The RMSE and CV values of the best fit are 0.5 mm h^−1^ and 20%, respectively. Best fits over individual TLPM (not shown) highlight small differences among them ($\Delta_{R}$ is less than 6%). On the other hand, there is a non-negligible departure from the ITU-R model. The latter returns smaller values of *R* at given γ values than the best fit based on local data. At 1 mm h^−1^, the difference is 40%, at 10 mm h^−1^ it is 20%, and it decreases below 10% beyond about 25 mm h^−1^. In this work, we used the best fit relationship in Figure 6 obtained from all the disdrometer data (same for all the events).

### 4.3. Comparison between CML and RG

Figure 7 reports an example of the basic steps of CML data processing and the comparison with RG+DIS data for the moderate rainfall event of 14–15 July 2019. In Figure 7a is the difference between TSL and RSL (i.e., raw data) for the four frequency channels of CML 11. In Figure 7b is total signal attenuation in the presence of rain, which has been obtained as the difference between the baseline level (BL), i.e., the RSL in clear-sky conditions, and the current RSL value. In Figure 7c is rain attenuation, i.e., the value in Figure 7b after subtraction of the attenuation component due to wet antennas, and Figure 7d shows the estimated rainfall depth. Non-zero values outside the time intervals identified as wet were dumped to zero. Finally Figure 7e has the rainfall depth measured by the four rainfall sensors surrounding CML 11. Please note in Figure 7a the difference in TSL-RSL among the channels even when it is not raining, that is, when the atmospheric attenuation across the channel is small. This means that such a difference is not only contributed by the propagation channel but also by the system itself (e.g., transmitting and receiving chain). When the RSL is subtracted from the BL, the four time series tend to overlap. The propagation path of CML 11 runs through the Valmalenco valley from SW to NE. There are two sensors (PRI and LAP) close to its terminals, while FUB and LAG are located midway through the path, on the mountain chests surrounding the valley. The time series obtained from CML data look very correlated with the occurrence of rain as detected by RG+DIS. Figure 7a,b shows that the RSL starts dropping (i.e., attenuation increases) when the RG+DIS detect rain, and the RSL returns to the initial level much after rain has ceased. This additional loss is much likely due to wet antennas. A wet antenna loss up to about 2 dB is visible during the initial transient.

Let us now assess the performance of CML as rainfall sensors over the entire dataset of events listed in previous Table 3. Rainfall occurrence and rainfall depth were evaluated over 10 min time intervals, i.e., the shortest available integration time, which is lower bounded by the RG dataset. The following Table 5 shows the occurrences of rain during the observation period, i.e., the number of 10 min slots flagged as wet according to the RG+DIS associated with every CML (column 3). The percentage of wet slots goes from 15 to 23% of the valid slots. After filtering out the wet slots corresponding to rainfall intensities less than $R_{min}$ (see Section 3.4), a majority of samples were discarded in the cases of CML 10, 11, and 13, along with most of them in the case of the 14 km link transmitting at Ku band. If we further discard highly inhomogeneous rainfall conditions setting a 0.5 threshold on the indicator $\Delta_{ext}$ in Equation (Equation 9), the population is drastically reduced. Indeed, rainfall in Valmalenco valley is very often patchy, especially in the Northern part of it, where orographic effects are important. Moreover, the paths of both CML 11 and 13 run between two mountain chests surrounding the valley, hence amplifying the above effects.

The contingency table shown as a barplot graph in Figure 8 is an indicator of how dry/wet slot classification based on CML data is good as compared with the classification based on RG+DIS. Figure 8a considers all the valid data (column 2 of Table 5), whereas in Figure 8b, only data corresponding to rainfall values above $R_{min}$ are retained, according to the procedure outlined in Section 3.4. True positives in the figure are wet slots as identified by both the CML-based algorithm and the associated RG+DIS. Results in Figure 8b are good: the sensitivity, i.e., the ability to correctly identify wet slots, is above 90%. The specificity, i.e., the ability to reject false negatives (i.e., dry slots), is close to 100% everywhere due to the filtering process.

Figure 9 shows the scatterplots of 10 min rainfall depth gathered from individual CML against the average of the RG+DIS set associated to each CML, calculated as in Section 3.4. The data points drawn in the figure include all the wet slots (column 3 of Table 5). On the other hand, best fits were obtained only from the data points above $R_{min}$ (column 4 of Table 5). The scatterplots highlight different patterns. CML 9 and 10 underestimate the rainfall depth with respect to RG+DIS. CML 9 is the longest link in the area; hence, the probability of inhomogeneous rainfall across its path is rather high. Even though the procedure proposed in Section 3.4 produces a RG+DIS-based rainfall estimate that approximates a path-averaged measurement, the few sensors available (six across a 14 km path) and their position (only three of them are within 1 km of its path) may explain the above discrepancy. CML 10 and 13 are shorter (8.4 and 7.0 km, respectively) and rather well covered by rainfall sensors (four and five, respectively). Even though for CML 13, the values of $\Delta_{ext}$ indicate the presence of patchy rainfall most of the time, there is very good agreement with RG+DIS. In the case of CML 10, there are fairly uniform rain conditions along the path during 141 time slots (that is about 24 rainy hours). However, if we restrict the best fit only to the above data, there is not a significant increase in the slope of the best fit line. This circumstance may suggest that the algorithm of rainfall depth estimate over a path based on RG+DIS works fairly well. CML 11 shows a very large overestimate, which is not easy to justify. The overestimate is rather independent of the rainfall depth; i.e., the coefficients of the best fit are similar when we select only specific rainfall depth intervals.

There are a few reasons that may explain the above discrepancies. First of all, it is known that rainfall exhibits very irregular patterns (both in time and in space) in a mountain climate. More accurate results would be probably achieved by using the *R*-γ best fits on an event basis, rather than an average relationship. Moreover, the algorithm for CML validation by RG+DIS in Section 3.4 could be improved by correcting RG and DIS measurements with the altitude to account for the vertical structure of rainfall. On the other hand, spatially inhomogeneous precipitation is not expected to produce large errors in the estimate of the average rainfall depth obtained by inversion of the non-linear Equation (Equation 1), at least in the frequency bands of the CML in Valmalenco (Ka and Ku bands), as the exponent α of the power-law best fit is very close to 1. We simulated this effect using simple patterns of uniform rainfall along a fraction of the link path and no-rainfall elsewhere. We got negligible differences between the average rainfall intensity over the path and the estimated average from inversion of Equation (Equation 1). Wet antenna attenuation was estimated by inspection of the RSL patterns, as shown in the example in Figure 7, rather than by a well-settled procedure, and we did not attempt to discriminate cases where only one of the terminals was affected. A bias in the estimate of wet antenna attenuation would affect rainfall intensity. In fact, by elemental mathematics, we get
(10)$$\hat{R}=R\left(1+\frac{1}{\alpha }\frac{\Delta }{A}\right)$$
where Δ is a (constant) error on the estimate of attenuation A=γL and $\hat{R}$ is the estimated rainfall intensity. Equation (Equation 10) holds in case Δ≪A. For instance, with α=1.1095 at 18.80 GHz, Δ=1 dB and A=5 dB, we get $\hat{R}=1.18R$. The percent difference between $\hat{R}$ and *R* decreases if *A* (i.e., *R*) increases. Finally, we can not rule out issues due to the CML network, such as errors or biases in power measurements, errors in RSL encoding (e.g., the available RSL value may differ from the instantaneous or quasi-instantaneous signal level), or again, errors on the metadata provided by the mobile company (e.g., the actual operational frequency of CML).

## 5. Conclusions

We assessed the performance of a network of CML when used as opportunistic rainfall sensors in a challenging mountainous environment located in Northern Italy. The benchmark rainfall data were direct measurements collected by an operational network of rain gauges and data from three disdrometers (laser precipitation monitors manufactured by Thies Clima) deployed in the study area. The particle counts returned by the disdrometers were processed to find the actual relationship between rain attenuation (i.e., the quantity measured by CML) and rainfall intensity, which is different from the one predicted by the global ITU-R model. The latter is usually adopted in the analysis of CML data.

The coarse quantization step of CML raw data (1 dB) does not allow one to measure light rain up to about 1–2 mm h^−1^, depending on CML characteristics. Moreover, it reduces the accuracy of rainfall intensity estimates based on nearly instantaneous raw data. In the worst case, the uncertainty of single-sample estimates is within 10% only if rainfall intensity exceeds 10 mm h^−1^. In this work the integration time is 10 min, corresponding to 60 CML raw data samples. Another limitation is the characteristics of rainfall in the measurement area, which is often patchy and intermittent. Hence, it is rather difficult to validate CML outcomes against direct rainfall measurements of rain gauges and disdrometers. To that end, we proposed a procedure that puts together rainfall estimates from the single-point sensors close to a CML to produce a rainfall depth value, which can be compared with the path-averaged measurement returned by a CML.

Once the occurrences of rainfall intensity below the minimum detectable signal are filtered out, CML exhibit good performance as rainfall detectors. On the other hand, their numbers as quantitative rainfall sensors over short integration times (10 min) are not as good, even when the sample rate of raw data is high (10 s in our case). Two of the four CML considered in this study have a significant bias in the rainfall depth (40% less and 70% in excess). It is important to state that, apart from the attenuation-to-rainfall intensity offline calibration through disdrometers, the CML used in this study are a set of fully independent sensors. In particular, the classification of wet and dry periods, and the calculation of the rain attenuation component, were carried out with no external data.

## Figures and Tables

**Figure 1 sensors-22-03218-f001:**
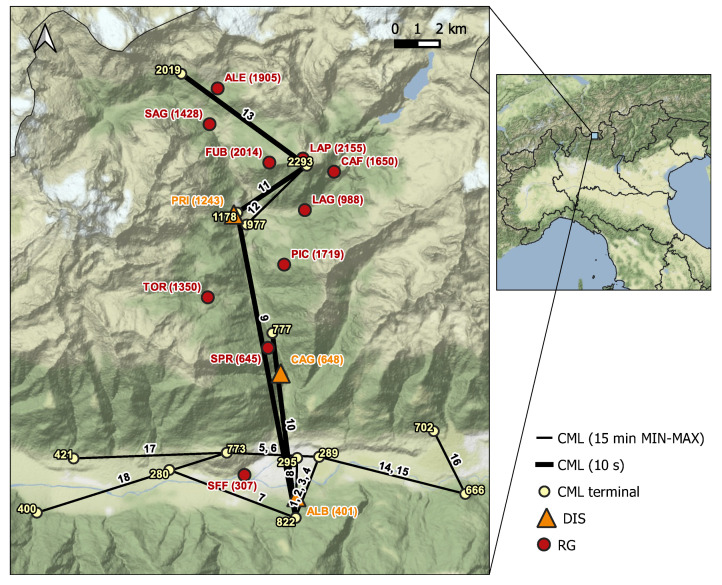
Sensors’ locations in the Valmalenco valley. CML are identified by numbers from 1 to 18, and DIS (orange) and RG (red) by three-character codes. In brackets are the altitudes (in m a.s.l.) at which sensors are located. The yellow numbers, instead, are the altitudes of CML terminals. The bold lines highlight the links for which nearly instantaneous power measurements (every 10 s) are available.

**Figure 2 sensors-22-03218-f002:**
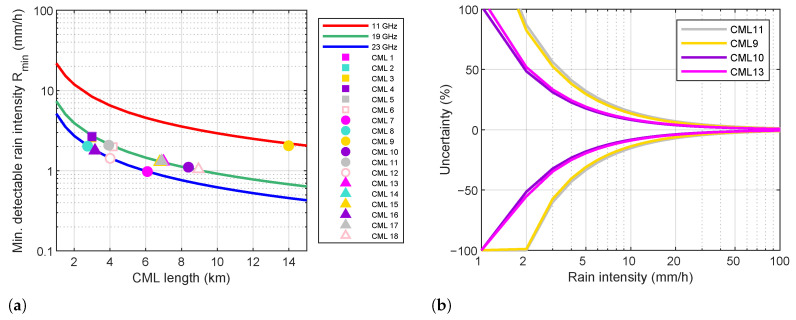
Sensitivity (**a**) and uncertainty (**b**) of rainfall intensity estimates due a 1 dB quantization error on CML raw data. The uncertainty plot in (**b**) shows the 95% confidence interval relative to the four CML used in this work to estimate rainfall intensity.

**Figure 3 sensors-22-03218-f003:**
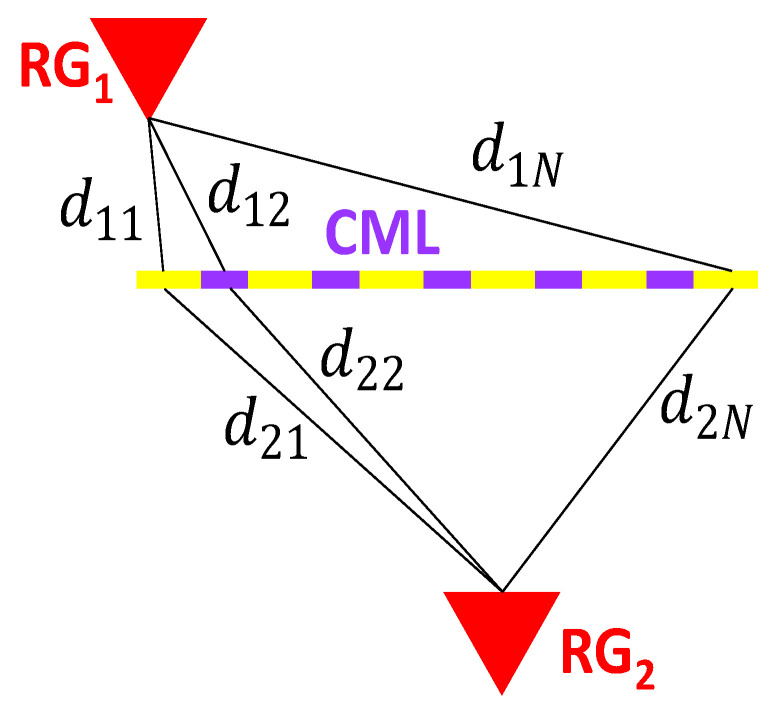
Effective distance of an RG from a CML.

**Figure 4 sensors-22-03218-f004:**
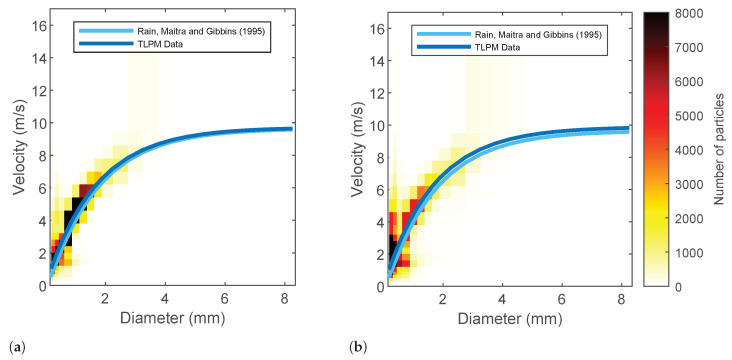
Relationship between raindrop velocity and raindrop diameter obtained from disdrometer data as compared with the Maitra and Gibbins model. (**a**) Moderate rainfall event (event 1 in Table 3), (**b**) heavy rainfall event (event 2 in Table 3).

**Figure 5 sensors-22-03218-f005:**
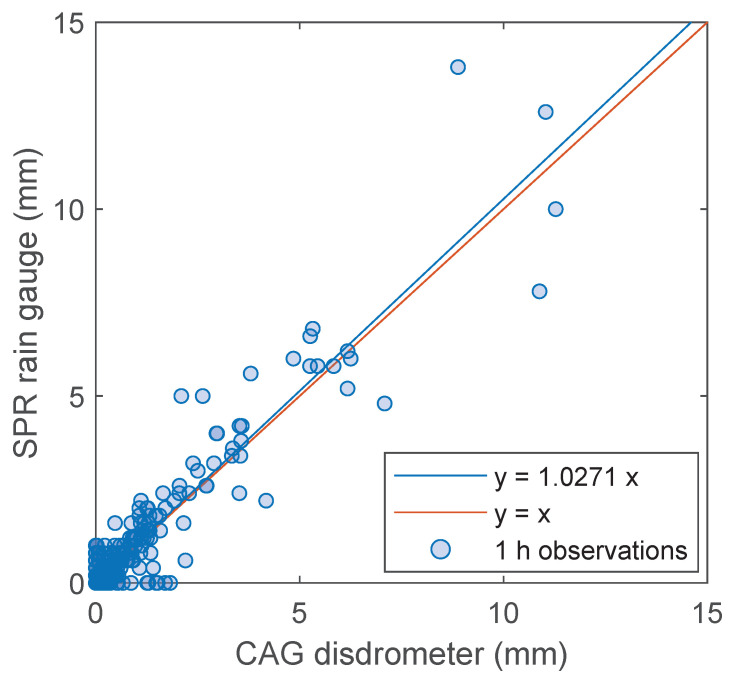
Scatterplot of one hour rainfall depths measured by the CAG disdrometer and the SPR rain gauge.

**Figure 6 sensors-22-03218-f006:**
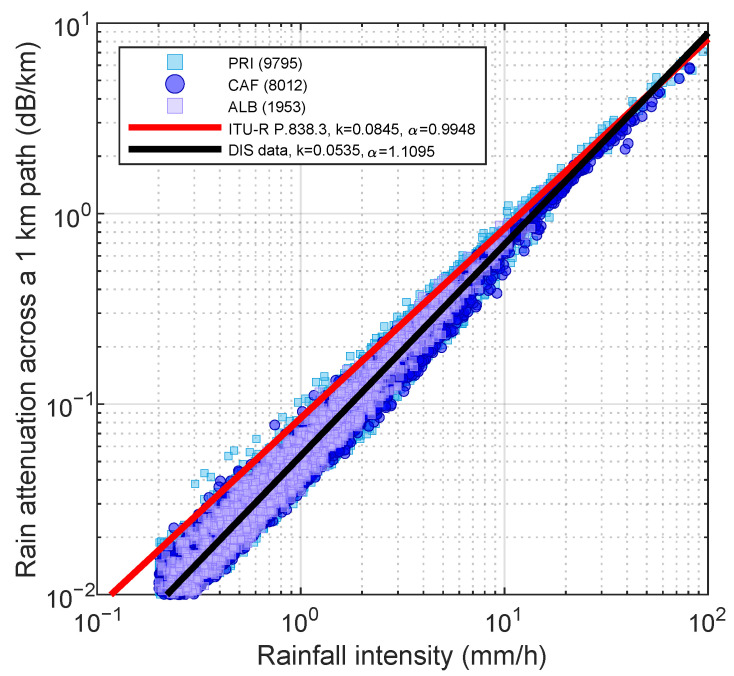
γ-*R* regression curve from disdrometer data relative to 15 rainfall events in Valmalenco (frequency: 18.8 GHz, polarization: V, direction of incidence: horizontal). In red is the ITU-R γ-*R* relationship. The numbers in brackets are the rainy minutes recorded by each disdrometer.

**Figure 7 sensors-22-03218-f007:**
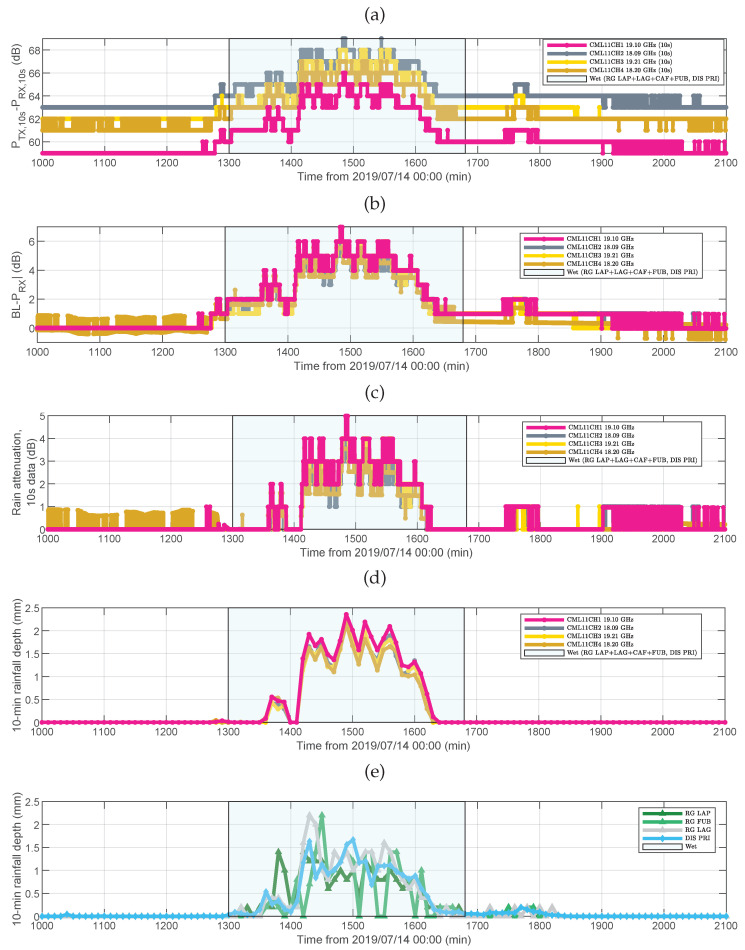
CML 11 and nearby rainfall sensors. (**a**) Difference between CML transmit and receive power levels (four channels), (**b**) total path attenuation assuming a zero-attenuation level with no rain, (**c**) rain attenuation, (**d**) 10 min rainfall depth, and (**e**) 10 min rainfall depth from nearby rainfall sensors.

**Figure 8 sensors-22-03218-f008:**
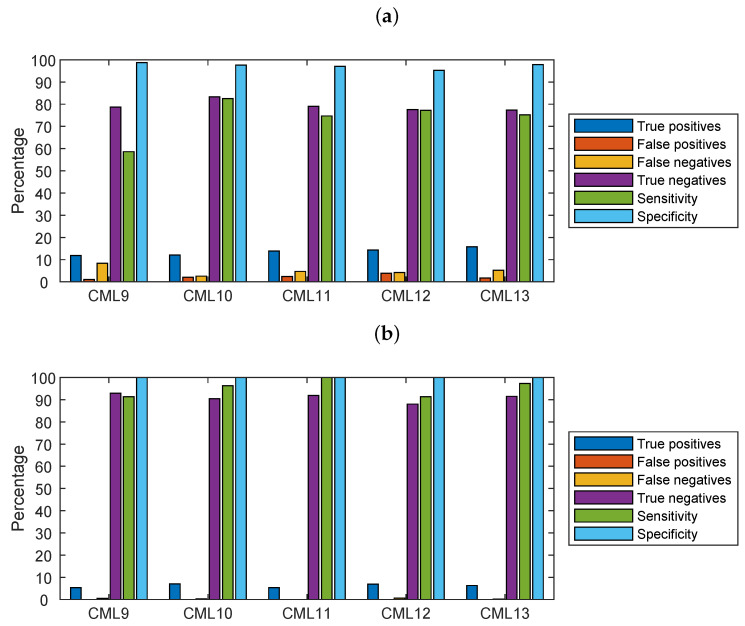
Contingency table, sensitivity, and specificity relative to the dry/wet classification algorithm based on CML: (**a**) all data, (**b**) only data with rainfall depth above $R_{min}$ (i.e., CML minimum detectable rainfall intensity).

**Figure 9 sensors-22-03218-f009:**
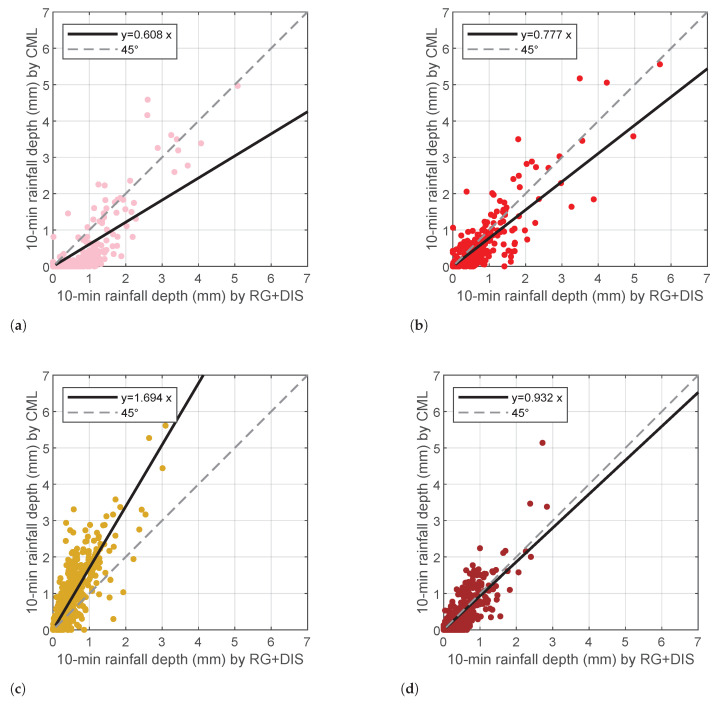
Scatterplots between CML-based and (RG+DIS)-based 10 min rainfall depth estimates: (**a**) CML 9, (**b**) CML 10, (**c**) CML 11, and (**d**) CML 13. Rainfall depths values below $R_{min}$ were filtered out before calculating the best fits.

**Table 1 sensors-22-03218-t001:** Characteristics of the 18 CML used in this study.

CML	Length (km)	Frequency (GHz)	Terminals Altitude (m a.s.l.)	Elevation (°)	Data Format
1	3.0	17.98, 18.09, 18.99, 19.10	289, 822	10.45	15 min MIN-MAX
2	3.0	18.03, 18.25, 19.04, 19.26	289, 822	10.45	15 min MIN-MAX
3	3.0	18.47, 18.58, 19.48, 19.59	289, 822	10.45	15 min MIN-MAX
4	3.0	18.19, 18.30, 19.20 , 19.31	289, 822	10.45	15 min MIN-MAX
5	4.2	18.03, 18.14, 19.04, 19.15	289, 773	6.71	15 min MIN-MAX
6	4.2	17.98, 18.09, 18.99, 19.10	289, 773	6.71	15 min MIN-MAX
7	6.1	22.02, 23.03	280, 822	5.12	15 min MIN-MAX
8	2.7	22.02, 23.03	295, 822	11.27	15 min MIN-MAX
9	14	10.74, 10.82, 11.23, 11.31	822, 1176	1.46	10 s
10	8.4	17.76, 18.77	777, 822	0.31	10 s
11	3.8	18.09, 18.20, 19.10, 19.21	1178, 2293	17.21	10 s
12	3.8	22.02, 23.03	977, 2293	20.35	15 min MIN-MAX
13	7.0	17.76, 18.77	2019, 2293	2.25	10 s
14	6.8	17.98, 18.09, 18.99, 19.10	289, 666	3.19	15 min MIN-MAX
15	6.8	18.03, 18.14, 19.04, 19.15	289, 666	3.19	15 min MIN-MAX
16	3.2	22.02, 23.03	666, 702	0.65	15 min MIN-MAX
17	6.9	17.76, 18.77	421, 773	2.93	15 min MIN-MAX
18	9.0	17.76, 18.77	421, 773	2.39	15 min MIN-MAX

**Table 2 sensors-22-03218-t002:** Rules for classification of 15 min time slots into dry and wet according to the values of the indicator in (Equation 4).

$W_{i,1}$	$W_{i,2}$	Outcome
$W_{i,1}\geq 0.5$	$W_{i,2}\geq 0.5$	Wet
$W_{i,1}\geq 0.5$	$0<W_{i,2}<0.5$	Wet
$W_{i,1}\geq 0.5$	$W_{i,2}=0$	Uncertain
$W_{i,1}\geq 0.5$	Not available	Wet
$0<W_{i,1}<0.5$	$W_{i,2}\geq 0.5$	Wet
$0\leq W_{i,1}<0.5$	$0\leq W_{i,2}<0.5$	Dry
$0<W_{i,1}\leq 0.5$	Not available	Uncertain
$W_{i,1}=0$	$W_{i,2}\geq 0.5$	Uncertain
$W_{i,1}=0$	Not available	Dry
Not available	$W_{i,2}\geq 0.5$	Wet
Not available	$0<W_{i,2}<0.5$	Uncertain
Not available	$W_{i,2}=0$	Dry
Not available	Not available	Not available

**Table 3 sensors-22-03218-t003:** Database with the information of the 15 analyzed events.

ID	Start Time (UTC+1)	End Time (UTC+1)	No of Episodes	Rainy Time (min)	Min–Max Rainfall Depth RG+DIS (mm)	Max Rainfall Intensity DIS (mm h^−1^)
1	14 Jul 2019	15 Jul 2019	1	500	15–31	17
2	25 Jul 2019	26 Jul 2019	5	500	8–85	118
3	1 Aug 2019	2 Aug 2019	4	440	7–15	17
4	6 Aug 2019	7 Aug 2019	5	770	25–48	45
5	11 Aug 2019	13 Aug 2019	5	380	8–20	83
6	18 Aug 2019	22 Aug 2019	8	1120	50–68	122
7	25 Aug 2019	26 Aug 2019	3	260	2–24	42
8	30 Aug 2019	2 Sep 2019	3	210	5–15	15
9	5 Sep 2019	8 Sep 2019	6	1490	26–57	22
10	22 Sep 2019	23 Sep 2019	2	570	13–24	22
11	1 Oct 2019	2 Oct 2019	3	310	10–18	59
12	6 Oct 2019	7 Oct 2019	1	290	1–7	19
13	9 Oct 2019	9 Oct 2019	1	230	2–9	13
14	15 Oct 2019	16 Oct 2019	6	260	11–36	15
15	19 Oct 2019	24 Oct 2019	12	3900	54–176	118

**Table 4 sensors-22-03218-t004:** Performance of the power-law fit to the γ-*R* relationship, as estimated through disdrometer data (CAG, ALB and PRI). The indicators CV and $\Delta_{R}$ are defined in Equations (Equation 6) and (Equation 7), respectively. Frequency: 18.8 GHz, polarization: V, horizontal path.

Event ID	RMSE (mm h^−1^)			CV (%)			$\Delta_{R}$ (%)
	PRI	CAG	ALB	PRI	CAG	ALB	
1	0.5	0.3	-	19	13	-	8
2	1.9	1.6	-	29	27	-	15
3	0.4	0.1	0.2	21	12	12	14
4	0.5	0.3	0.3	14	14	13	11
5	0.7	0.8	-	15	19	-	7
6	0.7	0.5	-	19	10	-	7
7	0.5	0.5	-	17	8	-	22
8	0.3	0.1	-	19	12	-	16
9	0.3	0.3	-	11	10	-	17
10	0.5	0.5	-	35	32	-	9
11	0.5	0.7	-	18	17	-	11
12	0.2	0.1	0.2	17	12	20	44
13	0.6	0.3	0.4	32	23	29	4
14	0.4	0.3	-	16	16	-	21
15	0.4	0.2	0.1	19	12	20	13

**Table 5 sensors-22-03218-t005:** Population of 10 min time slots in the observation period (6063 slots) and their classification according to the set of RG+DIS associated with each CML.

CML	Valid	Wet	Wet + above $R_{\min}$	Wet + $\Delta_{\mathrm{ext}}<0.5$	Wet + above $R_{\min}$ + $\Delta_{\mathrm{ext}}<0.5$
9	5907	1201	300	30	25
10	5911	867	409	182	141
11	5894	1104	278	19	16
13	4743	1103	406	15	12

## Data Availability

Third party data. Data were obtained from Vodafone Italia S.p.A. and are available with the permission of Vodafone Italia S.p.A.

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
