# Peer review of "Comparison of CML Rainfall Data against Rain Gauges and Disdrometers in a Mountainous Environment"

_sensors, 2022, doi:10.3390/s22093218_

Round 1

Reviewer 1 Report

In this paper, a rainfall observation platform based on CML is presented in a mountainous environment in northern Italy, and related inversion processes are described (e.g. dry/wet differentiation, baseline determination, wet antenna attenuation correction, etc.) in order to demonstrate the performance of rainfall observation in a challenging mountainous environment based on CML. There are a total of 18 links (frequency in the teens to twenties GHz, length from 2.7 km to 14 km), four of which are in the format of “10s” and are used to retrieve rain intensity, while the other 14 are in the format of “15 min MIN-MAX” and are only used to distinguish dry and wet period. In addition, data from 3 TLPM disdrometers and 10 rain gauges are used as truth reference. And it turns out, “CML shows a good performance in detecting precipitation, whereas quantitative rainfall estimates may have large discrepancies”.

Overall, the study is interesting for CML rainfall inversion fields mentioned as an advantage: they can work in mountainous areas, which weather radar cannot. In addition, the author's research is very detailed, and every link of the inversion process is explained and discussed in detail. In the result part, a reasonable explanation is given for the error. However, it should be mentioned that there are a lot of inconsistencies and writing errors in the paper. Moreover, there are some cases of repeated descriptions, such as Sections 3 and 4 interspersed with each other, which makes reading difficult, I suggest the author improve his presentation. Overall above situation I give the opinion is minor revision.

Below are some specific comments.

Major comments:

  1. Line 115: There is a serious physical problem here. The extinction cross section in equation (4) is actually related to the incident direction of electromagnetic wave for flat ellipsoid. This may not have a serious impact on the near-horizontal path, but the inclination of the link here is up to 20 degrees.
  2. Line 235: How to set Tg rule?
  3. Line 240: What are the specific parameter Settings for antenna wetting correction?
  4. Line 353: “We did not consider events with snow or solid precipitation”. How did you do that?
  5. Figure 7: Why did the rain attenuation recur between 1900 and 2100?

Minor comments:

  1. Line 10: Is it appropriate to say “a CML”, where CML is defined as “Commercial Microwave Links”? There's more than one, not all of them.
  2. Line 15: The Keywords “Rain fading” is not appropriate, which doesn't appear in the text.
  3. Line 21: There are different expressions such as "rain rate", "rainfall rate", "rain intensity" and "rainfall intensity". Please use the same expression to avoid misunderstanding. There's more than one, not all of them.
  4. Line 28: There are different expressions such as "rain attenuation", "attenuation", "rain intensity" and "rainfall intensity". Please use the same expression to avoid misunderstanding. There's more than one, not all of them.
  5. Line 31: There are different expressions such as "drop size distribution", "raindrop size distribution", "rain intensity" and "rainfall intensity". Please use the same expression to avoid misunderstanding. There's more than one, not all of them.
  6. Line 75: There are different expressions such as "rainfall sensors" and "rain sensors". Please use the same expression to avoid misunderstanding. There's more than one, not all of them.
  7. Line 80: There are different expressions such as "rainfall" and "precipitation". Please use the same expression to avoid misunderstanding. There's more than one, not all of them.
  8. Line 94: There are different expressions such as "rain depth" and " rainfall depth". Please use the same expression to avoid misunderstanding. There's more than one, not all of them.
  9. Line 97: Roman numerals should not be used here.
  10. Line 100: Please capitalize "Section".
  11. Line 100: “15 rainfall events”?
  12. Line 126: There are different expressions such as " raindrop diameter" and "drop diameter".
  13. Line 132: There are different expressions such as "fall speed" and "Velocity".
  14. Line 176: Change “sec.2.1” to “Section 2.1”. There's more than one, not all of them.
  15. Line 181: “available available”?
  16. Line 181: Incorrect format of citation.
  17. Equation (4): “γd”?
  18. Line 252: “(2)”?
  19. Line: “10 holds”?
  20. Line 507: “CMsL”?

Reviewer 2 Report

Manuscript ID: sensors-1660795
Title: Comparison of CML rainfall data against rain gauges and disdrometers in a mountainous environment.
Journal: Sensors
OVERVIEW
The authors assess the performance of a network of CML as opportunistic rainfall sensors in the challenging mountainous environment of Valmalenco valley, in Northern Italy.
1. The subject matter is actual, interesting and within the scope of the Journal Sensors.
2. The title fully describes the manuscript. 
3. The English is very good.
4. The structure of the manuscript is appropriate, and the research is well designed.
5. The results are consistent with the related bibliography. 
6. As for the rest, I have a few small revisions to suggest.
In conclusion, I believe this manuscript is worthy of publication after a minor revision.
Please read the specific comments.
SPECIFIC COMMENTS
Line 48: why divide by 1.14? Please justify.
Figure 1: Figure 1 has black lines with different widths. Please include the bold black line in the legend.
Figure 6b: The CML data only has acceptable accuracy for I > 10 mm/h. In conclusions (line 408) read “does not allow to measure light rain (up to about 1-2 mm/h)”. From Figure 6b, for I = 1 mm/h the uncertainty is 100% and for I = 2 mm/h the uncertainty is 50%. Please justify why the limit acceptable for the uncertainty is 50%. In my opinion, should be about 10% and in conclusions should read “does not allow to measure light rain (up to about 10 mm/h)”. Please justify.
In mountain climate, the rainfall is poorly distributed in space and time. In most cases, local effects may be correlated with altitude. However, seems that the distances of RG to CML were measured on a horizontal plane. Please explain why the beam hypsometry and rainfall gauge altitude were not used to correlate rainfall data?     
